# Effect of Dietary Protein and Processing on Gut Microbiota—A Systematic Review

**DOI:** 10.3390/nu14030453

**Published:** 2022-01-20

**Authors:** Shujian Wu, Zuhaib F. Bhat, Rochelle S. Gounder, Isam A. Mohamed Ahmed, Fahad Y. Al-Juhaimi, Yu Ding, Alaa E. -D. A. Bekhit

**Affiliations:** 1Institute of Microbiology, Guangdong Academy of Sciences, Guangzhou 510070, China; sjwu@stu2018.jnu.edu.cn; 2State Key Laboratory of Applied Microbiology Southern China, Guangzhou 510070, China; 3Key Laboratory of Agricultural Microbiomics and Precision Application, Ministry of Agriculture and Rural Affairs, Guangzhou 510070, China; 4Guangdong Provincial Key Laboratory of Microbial Safety and Health, Guangzhou 510070, China; 5Department of Food Science and Technology, Institute of Food Safety and Nutrition, College of Science & Engineering, Jinan University, Guangzhou 510632, China; 6Division of Livestock Products Technology, Sher-e-Kashmir University of Agricultural Sciences & Technology of Jammu, Jammu 180009, India; zuhaibbhatvet@gmail.com; 7Department of Food Sciences, University of Otago, Dunedin 9016, New Zealand; rochelleshaguna@yahoo.com; 8Department of Food Science and Nutrition, College of Food and Agricultural Sciences, King Saud University, Riyadh 11451, Saudi Arabia; iali@KSU.EDU.SA (I.A.M.A.); faljuhaimi@ksu.edu.sa (F.Y.A.-J.)

**Keywords:** dietary protein, processing, gut microbiota, meta-analysis, influence, health

## Abstract

The effect of diet on the composition of gut microbiota and the consequent impact on disease risk have been of expanding interest. The present review focuses on current insights of changes associated with dietary protein-induced gut microbial populations and examines their potential roles in the metabolism, health, and disease of animals. Preferred Reporting Items for Systematic Reviews and Meta-Analysis (PRISMA) protocol was used, and 29 highly relevant articles were obtained, which included 6 mouse studies, 7 pig studies, 15 rat studies, and 1 in vitro study. Analysis of these studies indicated that several factors, such as protein source, protein content, dietary composition (such as carbohydrate content), glycation of protein, processing factors, and protein oxidation, affect the digestibility and bioavailability of dietary proteins. These factors can influence protein fermentation, absorption, and functional properties in the gut and, consequently, impact the composition of gut microbiota and affect human health. While gut microbiota can release metabolites that can affect host physiology either positively or negatively, the selection of quality of protein and suitable food processing conditions are important to have a positive effect of dietary protein on gut microbiota and human health.

## 1. Introduction

Recent research provides strong evidence that gut microbiota plays an essential role in human health [1,2]. In parallel, increasing knowledge about the impact of diet on the composition of the gut microbial population is emerging, which consequently impacts human health and disease [3]. For example, a relationship has been reported for human immune status [4], neurodegenerative diseases [5], metabolic syndrome [6], and so on. Gut microbiota metabolizes dietary components and releases metabolites that influence host physiology either positively or negatively [7]. The complex relationship between diet, gut microbiota, and human health has gained tremendous interest as a natural system to improve health and wellbeing [1,2].

Among the various food nutrients, protein has received increasing attention since it is the primary substrate for important beneficial short-chain fatty acids (SCFAs) and harmful putrefactive metabolites (such as ammonia, amines, hydrogen sulfides, phenols, and indoles), which can be produced by gut microbiota through proteolytic fermentation and may influence host health and contribute to the risk of diseases [7,8,9,10]. Some of these metabolites have bioactive properties and play a critical role in signaling and gene expression of the host [11] and have potential involvement with health issues such as cardiovascular and metabolic diseases [10]. Protein intake in terms of quantity and quality is central to the above-mentioned effects, with complex mechanisms involved. For example, high protein intake was found to be associated with an increased risk of inflammatory bowel diseases [12], and, at the same time, it can induce satiety through increased production of the anorectic hormone peptide YY [13]. However, information available about the influence of the amount and source of dietary protein on microbiota, host health, and metabolism has not been critically evaluated. The ingested dietary protein may change both the diversity and composition of the gut microbiota [11], and, thus, the topic deserves an updated review. It is advantageous to examine dietary protein-induced changes on gut microbiota and to understand the corresponding metabolic functions and how protein source and its processing can affect the relative abundances of microbial populations and their influence on host physiology. This systematic review aims to delineate the impact of various dietary protein levels and dietary protein sources and their processing on relative abundances of gut microbial population and examine potential underlying factors that may influence this relationship. The scope of this review is to examine the physiological functions associated with various gut microbial populations that are influenced by dietary proteins and the processing methods, as reported in humans, animals, and in vitro models, and provide insights on their potential roles in metabolism, health, and disease.

## 2. Materials and Methods

The guidelines of Preferred Reporting Items for Systematic Reviews and Meta-Analysis (PRISMA) protocol [14], with a prespecified search strategy, eligibility criteria, extraction process, and objectives, were used to perform this systematic review.

### 2.1. Literature Search

Electronic databases, including PubMed, Scopus, Web of Science (core collection), and Central (Cochrane central register of controlled trials), were chosen to collect related literature from 2011 to 2021 by using a combination of subject headings, free-text terms, synonyms, and key words relevant to this review. The search terms included “dietary protein” OR “protein intake” OR “protein consumption” OR “protein metabolism” OR “protein digestion” OR “protein fermentation” AND (microbiota OR microbiome OR microflora OR commensal OR bacteria* OR microbial) AND (gut OR gastrointestinal OR intestine* OR “digestive tract” OR enteric OR duoden* OR jejun* OR ileum OR ileal OR caec* OR cec* OR colon OR colonic OR fecal OR faecal. Asterisks were used to include any derivatives of keywords. We tried to ensure the inclusion of the most recent studies and the most comprehensive search up to 2021.

### 2.2. Study Selection Criteria

In the initial research, records attained from the search results were merged into the reference management software Endnote (X9 version) and de-duplicated prior to screening the abstracts for relevance. Studies that investigated the effects of protein source/dose modification on the gut microbiota were deemed as potentially relevant, and other records were deemed irrelevant. Full-text articles of the relevant studies were reviewed and assessed for eligibility according to prespecified eligibility criteria. Studies were included if they met all the following criteria: (1) performed experimental research (dietary interventions/treatments) on healthy humans, mice, rats, pigs, or in vitro; (2) dietary interventions or experimental research with protein modification was the primary aim; (3) dietary interventions that administered normal or high protein doses or increasing levels of protein; (4) presence of control; and (5) measured abundances or quantified gut microbial populations.

### 2.3. Data Extraction

Supplementary data or supporting information were also referred to during the extraction of relevant data from studies using a predesigned form. The extracted data can be summarized as: (1) model characteristics (type, breed, and male or female); (2) the type of sample analyzed for gut microbiota (feces, cecal contents, colonic contents, ileal contents, and cecal mucus); (3) dietary protein characteristics (type and dose); (4) impact of dietary protein on gut microbiota (i.e., changes of increase or decrease) in gut microbial population induced by protein sources compared to the control. Gut microbiota changes that were not reported in text were derived from reported figures and tables of relative abundances or concentrations of various bacterial populations. If changes in microbial populations were not significant, the increasing and decreasing trends of various bacteria were recorded. If there were no differences in abundances of bacteria compared to the control, it was not recorded unless it was the most abundant phylum, family, or genus in the control or if the control had the highest/lowest abundance of the particular population compared to other diet groups in the study.

Specific details regarding the indication of abundances are given below:(a)If the most abundant population in that diet group, one asterisk was placed after classification (*);(b)If the most or least abundant population in that diet group compared to all other diet groups, two asterisks were placed after classification (**);(c)If the most abundant population in that diet group, as well as compared to other diet groups, three asterisks were placed after classification (***);(d)If least abundant compared to other diet groups but most abundant in that diet group, four asterisks were placed after classification (****);(e)If there are no differences in the number of asterisks for a particular bacterial population between diet groups, then there are no differences in the abundance of the corresponding microbial population amongst those diet groups.

## 3. Results

### 3.1. Selected Studies

The process of the literature selection is shown in Figure 1. A total of 1390 relevant records were identified and selected by the literature search from the following electronic databases, and the retrieved studies are shown in brackets: PubMed (*n* = 59), Scopus (*n* = 755), Web of Science (core collection) (*n* = 441), Central (Cochrane central register of controlled trials) (*n* = 114), and from the manual-searching of reference lists (*n* = 21). A total of 407 duplicate records were removed by the de-duplication process, and the remaining 983 records were evaluated for relevance, which subsequently resulted in 828 records that were deemed irrelevant according to the study selection criteria. We excluded studies that were conducted on other animals (e.g., cats, dogs, marine animals) and insects (48), studies investigated dietary interventions on unhealthy participants with an acute or chronic disease/condition (14), studies reporting on protein effects not being the primary aim (35), studies which focused on low protein or protein-deficient/restricted diets (23), and studies that implemented ineligible control (fiber) and experimental designs (6). A total of 29 studies were found to meet all the set criteria and were selected to be included in this systematic review. These studies included 6 mouse studies, 7 pig studies, 15 rat studies, and 1 in vitro study, which are discussed in detail in the subsequent sections. The bibliometric information of these 29 studies can be found in Table 1.

### 3.2. Effect of Protein Source on Gut Microbiota

Proteins are widely found in nature, and their nutritional, structural, and functional properties vary dramatically [43,44]. These aspects of proteins depend on the amino acid sequence, the amino acid type, the polypeptide charge, and the three-dimensional arrangement of the polypeptide structure [45]. Adult humans with sedentary lifestyles need a minimum of 0.8 g of protein/kg body weight, whereas slightly higher amounts are recommended for more active individuals [46]. A wide range of protein sources are used for food (e.g., animal-based, marine-based, plant-based, and insects, as well as single cells such as yeast and algae) depending on their availability, morals, religious permissibility, and affordability. This diverse range of proteins results in the generation of different peptides in the gastrointestinal tract and different kinetics of protein digestion [19]. Peptides and amino acids generated from the digestion of dietary proteins influence the composition of gut microbiota [47,48] (Figure 2). Foods can also become indigestible as a result of chemical modifications such as oxidation and remain unaffected by gastric digestion due to structural configuration and reach the large intestine in a complex form containing various macromolecules, including proteins [26,30,49]. These proteins in the large intestine are used as a substrate for microbial fermentation and putrefaction processes [7] and shape the diversity of gut microbiota [26,35,36,50] (Figure 2).

The status of the proteins (i.e., level of post-translational modifications caused by food processing/preparation, gastric digestion, and protein interaction with other food components) can modify the composition of gut microbiota and lead to the production of different peptides during digestion, which could reciprocally affect gut microbiota [47]. For instance, the relative abundance of *Akkermansia* increased in rats fed on hen egg white, whereas higher relative abundance of *Proteobacteria* and *Peptostreptococcaceae* and lower relative abundance of *Lachnospiraceae* were found in rats fed on duck egg white [29]. The observed differences were attributed to differences in peptide profiles that were produced during digestion. It has been reported that several microbial species such as *Bacteroidetes*, *Actinobacteria*, *Firmicutes*, *Proteobacteria*, *Roseburia*, *Lactobacillus*, and *Verrucomicrobia* are sensitive to peptides that result in changes in the composition and diversity of gut microbiota [47]. In addition, growing evidence has indicated that amino acids, products of dietary protein digestion, can affect the structure, composition, and functionality of gut microbiota [48,51]. The amino acids can further be metabolized into different microbial metabolites by gut microbiota, such as SCFAs, polyamine, hydrogen sulfate, phenol, and indole, and the resultant metabolites can be involved in various physiological functions that are related to host health and diseases, [48]. For example, an increase in the abundances of *Escherichia-Shigella*, *Aquabacterium*, and *Candidatus Methylomirabilis* and a decrease in the abundances of *Bacteroides*, *Bacillus*, *Pasteurella*, *Clostridium sensu stricto*, *Faecalibacterium*, *Paucisalibacillus*, and *Lachnoclostridium* were found in pigs with dietary lysine restriction (30%), which resulted in restricted amino acid metabolism [52]. The role of amino acids in regulating the host health was supported in various studies. For example, an increase in SCFA-producing bacteria (*Bifidobacterium*, *Lactobacillus*, *Bacteroides*, *Roseburia*, *Coprococcus*, and *Ruminococcus*) and inflammation-inhibiting bacteria (*Oscillospira* and *Corynebacterium*), as well as a decrease of inflammation-causing bacteria (*Desulfovibrio*), were observed in mice with methionine-restricted diets, which can collectively improve gut health [53]. Further, the amount of undigested protein flow to the colon depends on the intake level and digestibility of proteins from different food sources [54], which may affect the composition of gut microbiota involved in protein fermentation (Table 1). For example, piglets fed with highly digestible casein-based diets showed a higher count of *Enterobacteriaceae* than piglets fed on less digestible soybean meal-based diets [35] (Table 1). More fermentable proteins were found in the hindgut of piglets fed on Palbio 50 RD (P50, an animal protein source) that has low digestibility compared to concentrated degossypolized cottonseed proteins (CDCP) [36,55]. This led to a higher abundance of *Escherichia* (potential pathogenic bacteria) in the Palbio 50 RD-fed piglets compared to a higher abundance of *Lactobacillus* (beneficial bacteria) in the CDCP-fed piglets [36] (Table 1). The CDCP diet resulted in a higher intestinal accumulation of valeric acid and branched chain fatty acid concentrations, whereas the Palbio 50 RD diet resulted in higher ammonia, nitrogen, and methylamine contents. These findings support the differential effects of various proteins. This may be related to the reported digestibility differences among proteins. For example, studies have demonstrated that casein and whey proteins have different digestion kinetics (based on leucine kinetics modification) [56], beef and chicken proteins have a higher digestion rate (digestibility) than fish proteins [57], and soy proteins to have higher digestion kinetics than milk proteins (based on nitrogen absorption, splanchnic uptake, and metabolism) [58]. These observable changes in digestion parameters (leucine kinetics, digestibility, and nitrogen kinetics) can eventually manipulate gut health and, subsequently, the overall health of the host. For example, the abundance of Bacteroidales family S24-7 was enhanced in mice fed with soybean meal as the protein source compared to casein, spray-dried plasma protein, yellow meal worm, partially delactosed whey powder, or wheat gluten meal [19] (Table 1). Similarly, Zhu and co-workers, who conducted a series of studies using a rat model, found that the composition of gut microbiota was sensitive to the protein source [25,26,28]. These studies showed that muscle and plant foods had different effects on the growth of different microbial populations. For example, *Firmicutes*, a phylum that includes many pathogenic classes such as *clostridia* and *bacilli*, were increased in rats fed with proteins from beef, pork, or fish. *Bacteroidetes*, a phylum that contains a large number of microorganisms involved in metabolic processes involving the hydrolysis of polysaccharides and proteins, were increased in rats fed with soy protein and decreased in rats fed with proteins from fish (Table 1). Surprisingly, *Fusobacteria*, microorganisms that are suspected to be involved in colon cancer [59], were decreased in rats fed with proteins from beef, pork, and fish (Table 1). In agreement with the above-mentioned studies, An et al. (2014) [23] found higher contents of *n*-butyric acid, lactic acid, and other putrefaction compounds in rats fed on soy protein and fish meal compared to casein, which suggested differential metabolism by gut microbiota that can lead to physiological changes in the gut. Collectively, results from these studies indicate that the source of dietary proteins can shape the composition of gut microbiota [19,23,60] (Table 1).

### 3.3. Animal and Plant Proteins

Dietary proteins from animal sources and plant sources have been widely explored in relation to the modulation of gut microbiome. Many studies have reported that plant proteins (e.g., rice, soy, wheat) can improve the composition of gut microbiota [19,61,62]. For instance, soybeans, an important source of plant proteins, have gained wide popularity due to their health-promoting effects [63]. Soy proteins are considered a rich source of all essential amino acids that preferentially support the growth of some gut microbiota as both nutrient and energy sources [62,63]. Soy proteins/peptides appear to modulate gut microbiota by exerting probiotic effects by enhancing probiotics (*Lactobacilli* and *Bifidobacteria*) and decreasing Bacteroidetes [62,63]. Han et al. reported that the diversity and richness of the gut microbiota in mice were changed by fermented soy whey, resulting in the enhancement of *Bifidobacterium*, *Lactobacillus*, *Butyricicoccus*, *Parabacteroides*, *Lachnospiraceae*, and *Akkermansia muciniphila* in affecting the metabolism and health of mice [64]. While there is a strong recent interest in advocating plant proteins as a healthier dietary option, several studies have highlighted the importance of animal proteins in the human diet [65,66]. The proteins from animal-based food sources may have better effects on gut microbiota compared to plant-based food sources due to the higher protein digestibility of animal proteins and the fact that the digestion of plant proteins may be limited by the presence of antinutritional factors found in plants [67]. Animal proteins have more balanced essential amino acids than plant proteins [68,69] and are thus considered higher quality protein. As discussed above, a lower abundance of *Fusobacteria* was found in rats fed on animal proteins (Table 1). Dairy and meat protein intake at a recommended level increased the abundance of the genus *Lactobacillus* and maintained a more balanced composition of gut microbiota compared to soy protein, which is beneficial to the host [25,26,28]. The counts of probiotic *Lactobacillus* and *Bifidobacterium* were higher in rats fed with casein than those fed with soya and zein proteins [22] (Table 1).

### 3.4. Effect of Protein Content and Diet Composition on Gut Microbiota

#### 3.4.1. Protein Content

Protein is an important source of essential nutrients and is necessary to maintain a normal body and health [70]. However, chronic consumption of improper protein intake can result in serious pathological and degenerative diseases involving gut microbiota [15,17,18]. The amount of undigested protein transferred to the large intestine increases with the increase of dietary protein intake [11]. Several metabolic pathways regulate proteolytic fermentation of protein (e.g., amino acid catabolism versus biosynthesis of microbial proteins) and produce diverse metabolites to influence the luminal environment [7] and gut microbiota [50]. The composition and metabolism of the gut microbiota can change with the change in colonic luminal environment and substrate availability. For instance, luminal pH, branched-chain fatty acids, and propionate production were increased in a validated in vitro gut model subjected to a high protein diet compared to a low protein diet, which contributed to a range of changes in the composition and activity of gut microbiota [71]. Several metabolites, such as phenols, indoles, amines, sulfides, and ammonia, are produced by the microbial metabolism of dietary proteins, all of which can exert an adverse impact on intestinal health [9,72].

Some studies have reported significant feedback in response to a high-protein diet. Bacterial metabolites (e.g., SCFAs lactate, succinate, and formate) were reported to increase significantly in colonic luminal contents in rats on high-protein diets and to a lower extent in the cecal luminal content, which was associated with an increase in substrate availability and a reduction in *Faecalibacterium prausnitzii* and *C. leptum* groups and *Clostridium coccoides* counts in both cecum and colon [15] (Table 1). It is well known that acetate and butyrate play a key role in inhibiting the growth of pathogens in the gut [73]. A high protein diet induces a reduction of both propionate- and butyrate-producing bacteria and thus the production of propionate and butyrate [17,18]. This may lead to a favorable environment for pathogenic bacteria. An increase in some disease-associated bacteria (such as *Escherichia/Shigella*, *Enterococcus*, and *Streptococcus*) and a decrease in beneficial bacteria (such as *Ruminococcus*, *Akkermansia*, and *Faecalibacterium prausnitzii*) were observed in rats fed with a high-protein diet [17,18] (Table 1). Compared to a standard protein diet (20% casein and/68% carbohydrate), a high protein diet (30% casein and 57% carbohydrate) intake in mice showed an increase in *Bacteroidetes*, *Bacteroidaceae*, *Parabacteroides*, and *Bacteroides* and a decrease in *Firmicutes*, *Lachnospiraceae*, *Ruminococcaceae*, *Enterococcus*, and *Oscillibacter* [9] (Table 1). High casein and whole milk protein intake in rats [15,17,18] showed an increase in *Bacteroidetes*, *Bacteroides*, *Parabacteroides*, *Enterococcus*, *Escherichia/Shigella*, *Lactococcus*, *Streptococcus*, *Lactobacillus delbrueckii subsp. bulgaricus*, *Lactobacillus murinus*, *Shigella flexneri*, and *Streptococcus hyointestinalis* and a decrease in *Firmicutes*, *Akkermansia*, *Prevotella*, *Roseburia*, *Ruminococcus*, *Akkermansia muciniphila*, *Bifidobacterium animalis*, *Faecalibacterium prausnitzii*, *Roseburia/Eubacterium rectale*, and *Ruminococcus bromii* (Table 1). The high abundance of *Bacteroidetes*, *Bacteroidaceae*, and *Bacteroides* spp. in high protein diets reflects an increase in proteolytic bacteria [74]. Although *Bacteroides* have been identified as primary carbohydrate degraders [75], some *Bacteroides* spp. are involved in amino acid fermentation [76] and are associated with high protein intake [35]. A high abundance of *Bacteroides* is undesirable as they are known clinical pathogens and exist in most anaerobic infections and have one of the strongest antibiotic resistance mechanisms amongst all anaerobic pathogens [75]. Additionally, high levels of *Bacteroides* and *Parabacteroides* have been associated with increased tumorigenesis [77] and colorectal cancer [78]. The increased abundances of *Enterococcus*, *Escherichia*/*Shigella*, *Streptococcus*, *Shigella flexneri*, and *Streptococcus hyointestinalis* indicate the potential enrichment of opportunistic pathogenic bacteria that may promote gut dysbiosis and increase the risk of severe gastrointestinal diseases when certain genetic or environmental conditions are modified in the host [74]. However, the increase in probiotics such as *Lactococcus* and *Lactobacillus* spp., which also exhibit proteolytic properties [79], can be considered a beneficial outcome and balance the increase in pathogenic bacteria. High protein diets can lead to a decreased abundance of carbohydrate utilizers belonging to *Lachnospiraceae*, *Ruminococcaceae*, *Akkermansia*, *Prevotella*, *Roseburia*, *Ruminococcus*, *Akkermansia muciniphila*, *Bifidobacterium animalis*, *Faecalibacterium prausnitzii*, *Roseburia/Eubacterium rectale*, and *Ruminococcus bromii* [74,80]. Clearly, this emphasizes the role of appropriate dietary carbohydrate intake in decreasing protein fermentation and maintaining the abundance of beneficial bacteria that produce favorable metabolites such as short-chain fatty acids [7]. Probiotics such as *Akkermansia muciniphila* play an important role in degrading mucin, which has been associated with host metabolism and immunity, as well as being a therapeutic target in multiple gastrointestinal, metabolic, immune, and cancerous diseases [81]. Hence, these results partially support the detrimental effects associated with high protein intake, especially in combination with low carbohydrate diets (Figure 3). Gut microbiota also appear to be negatively affected by low protein diets. For example, mice fed with a low protein diet showed a lower abundance of cecal *Roseburia* sp., *Alistipes* sp., and *Muribaculaceae* compared to mice with a standard protein diet [17]. *Alistipes* sp. could provide both beneficial and harmful effects [82] and has a strong association with a factor known as STAT3 (signal transducer and activator of transcription 3) and promotes its phosphorylation, which is required to activate STAT3 [83]. The STAT3 is a major inflammatory signaling pathway that maintains intestinal barrier homeostasis. Furthermore, a low protein diet led to a higher abundance of *Desulfovibrionaceae* (positively correlated with inflammation), which led to abdominal infections [84]. Thus, there appears to be an optimum protein intake that is required for normal gut health, and a higher or lower protein intake could have a negative impact on gut microbiota and, subsequently, on health (Figure 3). Clearly, that optimum balance is likely to vary among individuals depending on genetic background, physiological condition, gender, age, and race. This is an interesting research field that requires further investigation.

#### 3.4.2. Other Macronutrients

Food as a complex material can have various combinations of protein and other macronutrients that could have synergistic or antagonistic effects on gut microbiota. The diversity of gut microbiota is markedly affected by different dietary macronutrients [85], including proteins [47], fats [86], and carbohydrates [87] (Figure 3). Additionally, micronutrients such as polyphenols appear to play a similar role [88]. Thereby, the composition of gut microbiota can be manipulated by the synergistic action of different macronutrients in diet, especially the proportion of protein and carbohydrate. The carbohydrate content in diet influences the ratio of carbohydrate to protein reaching the large intestine and, consequently, the substrate fermented by bacteria [7,89]. Nakata et al. (2017) [90] reported that the carbohydrate–protein ratio in diet altered the composition of gut microbiota and the protein fermentation process, which inhibited indole production in the caecum. Rist et al. (2014) [35] found that the proliferation of *Bifidobacteria* was stimulated by the increase of nondigestible oligosaccharides in dietary proteins. Moreover, the indigestible complex plant material in the host gut can be decomposed through the actions of the *Lachnospiraceae* and *Ruminococcaceae* via carbohydrate-active enzymes, sugar transport mechanisms, and metabolic pathways [91]. The proportion of the family *Lachnospiraceae* and *Ruminococcaceae* was decreased, while the proportion of the genus *Bacteroides* and *Parabacteroides* was increased in mice fed with a high-protein and low-carbohydrate diet, which may result in a deleterious gut environment [9] (Table 1). The production of SCFAs from the fermentation of dietary fiber reduced the demand for amino acids as an energy source and led to a lower pH that inhibited microbial proteolytic enzymes (which work best at neutral pH), which subsequently suppressed protein fermentation and reduced the production of potentially undesirable methylated amino acid metabolites [7,76,92]. These effects support the contention that high fiber intake in diet may change protein fermentation pathways and could protect the host against inflammation and the disruption of cell cycles [7]. Thus, a high-protein/low-carbohydrate diet may cause a deleterious luminal environment that has harmful effects on intestinal health [9]. There is an ideal balance between protein and carbohydrate in diet for optimal microbial fermentation in the gut. Different metabolites result from various combinations of diet [9,90], which may cause the gut microbiota to adapt to the new gut environment. For instance, the fermentable polysaccharides in soybeans can decrease the production of putrefactive compounds (e.g., indole and ammonia) produced from soy protein [90]. Similar to macronutrients, the presence of polyphenols has been reported to influence the digestibility of proteins. While a positive impact of tea polyphenols was observed on the susceptibility of ovalbumin and lysozyme to peptic digestion at pH 1.2, a negative effect was observed on the hydrolysis of the proteins during pancreatin digestion at pH 7.5 [93]. In summary, a decrease in dietary protein level, less modified protein content, and the addition of different macronutrients are important nutritional requirements for a balanced gut microbiota system.

### 3.5. Effect of Processing Technologies on Dietary Protein to Influence Gut Microbiota

Food processing is an essential part of food production and preservation that is practiced worldwide. Various processing technologies, which can affect the properties and characteristics of proteins, can exert different effects on digestion kinetics of proteins and the release of beneficial and harmful metabolites upon protein catabolism by the gut microbiota. There are thermal and non-thermal processes technologies that are currently used during the preparation of food products by food manufacturers and consumers. These technologies may induce various levels of protein modifications (denaturation, aggregation, and oxidation) that can induce positive or negative influence on the digestibility, absorption, and functional properties of the dietary protein [31,94,95] and, subsequently, utilization by gut microbiota (Figure 4). Gut microbiota utilize amino acids derived from undigested dietary proteins as building blocks to assemble microbial cell components and ferment them as an energy source [96]. Proteins that evade enzymatic digestion undergo bacterial hydrolysis via the production of microbial extracellular proteases and peptidases, which results in free amino acids available for uptake by gut microbiota [97,98]. The process also generates a wide range of metabolites that could exert beneficial or harmful effects. The extent of proteolytic fermentation is mainly influenced by the source and amount of protein intake that escapes digestion and reaches the lower gut [7]. The source of dietary protein influences its digestibility and, consequently, the amount of undigested protein undergoing bacterial fermentation. Similar to the source of protein, processing technologies can manipulate the structure of the proteins and accessibility of digestive enzymes, which will affect the protein digestibility and control the amount of undigested protein available for bacterial fermentation. Shifting of gut microbiota towards increased protein fermentation influences the relative abundances of various microbial populations and gut dysbiosis [96]. Gut dysbiosis has been associated with numerous diseases such as inflammatory bowel disease, celiac disease, allergies, metabolic syndrome, asthma, obesity, and cardiovascular diseases.

In general, non-thermal processing technologies such as pulsed electric field, high pressure, and ultrasonication have been reported to improve the digestibility of food proteins such as muscle, milk, and egg proteins by inducing favorable alterations in the structure of food proteins and microstructure of food matrices [99,100,101]. However, intense and prolonged processing conditions can sometimes induce unfavorable changes in dietary proteins and food matrices. For example, ultrasonication oxidizes the free-SH groups to S–S bonds and causes a decrease in total SH content. It also induces structural transformations such as a conversion of α-helix to β-sheet, β-turn, and random coils and directly affects the surface hydrophobicity and availability of hydrolytic sites to digestive enzymes [102]. HPP affects the structure of proteins and may cause increased exposure of sulfhydryl groups that are highly susceptible to oxidation during digestion. Oxidation of these highly reactive free sulfhydryl groups is the most prominent mechanism for protein oxidation that leads to low food digestion [103]. Further, it can result in the formation of disulfide bonds, protein–protein interactions, or even aggregates and can decrease the digestibility or the rate of digestion by interfering with the hydrolysis of proteins by digestive enzymes. Disulfide content has been reported to have an inverse relationship with protein digestibility of different proteins [104,105]. Information on the effect of proteins processed with non-thermal emerging technologies on the composition of gut microbiota is generally lacking and needs immediate scientific attention.

Thermal processes can induce several physicochemical modifications, such as denaturation, thiol oxidation, loss of protein solubility, increased surface hydrophobicity, carbonylation, protein aggregation, and Schiff base formation, which can affect the digestibility of food proteins by modifying their characteristics [106]. Mild thermal processes such as sous-vide have been reported to have a positive effect on protein digestibility by inducing structural and conformational changes and the partial unfolding of proteins and exposing the hydrolytic sites to digestive enzymes [107]. However, intense thermal technologies such as stewing, steam cooking, and roasting can induce unfavorable modifications in the food proteins, such as increased disulfide (S-S) content, protein aggregation, severe oxidation, or cross-linking, which can negatively affect digestibility by limiting the access of proteases to active sites and thus leave partially hydrolyzed proteins in the gut. These partially hydrolyzed proteins or digestive end products are fermented by colonic flora and produce different mutagenic products [104]. Li et al. (2019) [36] examined the effects of steam dried fish on the gut microbiota of pigs and reported an increase in abundance of *Bacteroidetes*, *Bacteroides*, *Parabacteroides*, *Prevotella*, *Ruminococcus*, *Spirochaetes*, *Clostridium*, and *Escherichia* and a decrease in the abundance of *Firmicutes*, *Phascolarctobacterium*, and *Roseburia* compared to the other mildly processed protein sources (dried porcine mucosal tissue, concentrated degossypolized cottonseed protein, and enzyme-treated soybean meal) (Table 1). An increased *Bacteroidetes*-to-*Firmicutes* ratio is negatively correlated with metabolic disorders such as obesity and type 2 diabetes [108,109], thus entailing the potential role of steam dried fish in ameliorating the risk of metabolic disorders. High levels of *Bacteroides* and *Parabacteroides* have been associated with increased tumorigenesis [77] and colorectal cancer [78]. Increased levels of *Prevotella* have been associated with a compromised immune function [110], thus, denoting potentially detrimental gut microbiota changes. Additionally, the significant increase in potentially opportunistic pathogens belonging to *Spirochaetes*, *Clostridium*, and *Escherichia* further implies potentially adverse microbiota changes associated with steam dried fish consumption. Yang et al. (2018) [39] analyzed the effect of heating fish protein for 24 or 48 h on gut microbiota in an in vitro model of human distal colon. The heated fish protein exhibited a significantly increased abundance of *Dialister*, *Phascolarctobacterium*, *Dorea*, and *Intestinimonas.* A significantly increased abundance of *Enterococcus*, *Clostridium sensu stricto 1*, *Firmicutes*, *Streptococcus*, and *Arcobacter* was also observed in the 48-h-heated fish protein, whereas the abundance of *Parabacteroides* and *Clostridium sensu stricto 1* was significantly increased in the 24-h-heated fish protein compared to the unheated control. Furthermore, 24-h-heated fish protein showed a decrease in abundance of *Bacteroidetes* and *Firmicutes*, whereas 48-h-heated fish protein significantly decreased *Bacteroidetes* content (Table 1). The significantly high abundance of *Clostridium sensu stricto 1* in the heated fish proteins and its greater abundance in the 48-h-heated fish can be correlated to the high amount of undigested protein available as a substrate for fermentation [111] due to the formation of disulfide bonds upon thermal processing, which reduces protein digestibility [112]. *Arcobacter* is known to consist of potential zoonotic pathogens [113], and its highest abundance in the 48-h-heated fish-protein-treated samples further indicated the negative influence of the longer thermal processing of proteins. The higher *Fusobacteria*-to-*Firmicutes* ratio in the 24-h-heated fish-treated samples compared to a higher *Firmicutes*-to-*Fusobacteria* ratio in the 48-h-heated fish protein was attributed to the extended thermal processing period effects on protein structure and formation of chemical bonds. The differential changes in abundance of *Bacteroidetes* amongst the heated fish proteins were also associated with the thermal processing period. These changes emphasize the effect of protein thermal processing on the composition of microbiota. In a similar experiment, Han et al. (2018) [30] studied the effect of heated fish protein on the microbiota of rats compared to a control diet (casein). Compared to casein, heated fish protein increased abundances of *Collinsella*, *Ruminococcus gauvreauii*, *Actinobacteria*, and *Lactobacillus*. Additionally, heated fish protein decreased the abundances of *Proteobacteria*, *Bacteroidetes*, *Fusobacteria*, *Fusobacterium*, *Bacteroides*, and *Subdoligranulum.* Moreover, heated fish protein exhibited a significant increase in *Firmicutes* and *Ruminococcaceae_UCG-005* compared to casein. The decreased abundance of *Proteobacteria*, *Fusobacteria*, and *Bacteroidetes* in the heated fish proteins was correlated to a significantly higher abundance of *Actinobacteria*. A significant decrease in carbohydrate-metabolizing bacteria such as *Bacteroides* and *Subdoligranulum* [74,114] in heated fish proteins may be due to the increase in proteolytic bacteria, such as *Lactobacillus* [115], that can control other microorganisms effectively. Overall, the results stated above demonstrate the complex microbiota changes associated with processed fish proteins.

Xie et al. (2020) [20] studied the effects of thermal and nonthermal processing of pork on the microbiota of mice. The pork was processed in four different ways (emulsion-type sausage, dry-cured pork, stewed pork, and steam-cooked pork) and compared with mice fed on casein. All processed pork samples increased the abundance of *Bacteroidetes*, *Muribaculaceae-norank*, and *Faecalibaculum*. The abundance of *Firmicutes* increased in all processed pork samples except for the emulsion-type sausage, where it decreased (Table 1). The decreased abundance of *Firmicutes* in emulsion-type sausage protein was correlated to an increased abundance of *Actinobacteria*. *Actinobacteria* is known to accommodate genera such as *Bifidobacterium* that are involved in the degradation of complex carbohydrates such as cellulose [116], which may be present in the casings of the emulsion-type sausages. Additionally, an increase in *Actinobacteria* and *Lachnospiraceae-uncultured* was observed, with the highest abundance observed for emulsion-type sausage and dry-cured pork samples. *Lachnospiraceae-uncultured* has been closely correlated to the production of putrefactive metabolites upon protein fermentation, such as ammonium, *p*-cresol, and indole [74]. Thus, its high abundance in dry-cured pork protein suggests the possibility of reduced digestibility due to salting and drying, which may have altered the accessibility of protein cleavage sites by digestive proteolytic enzymes [112]. Increased abundance of *Proteobacteria* in stewed pork and steam-cooked pork suggests a possible increase in opportunistic pathogens belonging to this phylum [74]. The decreased abundances of *Akkermansia* belonging to *Verrucomicrobia* in stewed pork and steam-cooked pork entails negative gut microbiota changes as these groups accommodate *Akkermansia muciniphila*, which is known to exhibit benefits in host physiology by improving host metabolism, immunity [81], and the integrity of gut lining [117] as well as the alleviation of metabolic disorders such as colitis, diabetes, and obesity [118]. Lower abundances of these beneficial groups in the stewed pork compared to the steam-cooked pork suggests a poorer protein quality associated with stewed pork protein. Additionally, the authors [20] reported that proteins from stewed pork were least bioavailable among all samples, and this was attributed to the higher cooking temperature and longer cooking period (100 °C for 150 min) of stewed pork compared to the other processed pork samples (core temperature of 70 °C). Lower cooking temperatures reduce the formation of disulfide bonds, which results in less compact protein structures that can be digested easily [112]. However, higher cooking temperatures and longer cooking periods can aggregate proteins and increase the formation of disulfide bonds, resulting in a gel network that reduces protein digestibility. Processing methods also influence the fractions of peptides and amino acids released upon digestion [119], which can be associated with changes in surface hydrophobicity [120] due to protein aggregation and oxidation [121].

While the impact of traditional thermal processing methods on the composition of microbiota has been studied as discussed above, the effects of novel thermal technologies, such as microwave or ohmic heating, have not been evaluated yet. Microwave processing of food proteins at high temperatures induces several modifications, such as cross-link formation, changes in secondary structures and hydrophobicity, and the glycation of proteins through the Maillard reaction [122]. Microwaves can affect the secondary structure of proteins by disrupting the hydrogen bonds and exposing the hydrophobic regions by inducing the vibrations of polar groups and increasing the kinetic energy of protein molecules [123,124]. These changes can induce the aggregation of proteins and affect protein digestibility by compromising the efficacy of digestive enzymes and production and the release of digestive end products [125]. Similarly, ohmic heating, which applies the Joule effect to produce heat inside the food products through the application of alternating current [126], affects the structure of proteins and their interactions and the formation of protein aggregates [127]. The conformational, structural, and microstructural changes induced in the food proteins and matrices during ohmic heating are a result of a combined effect of thermal and non-thermal effects (electrochemical reactions and electrical effects) [127,128]. Ohmic heating has been reported to increase the digestibility of whey proteins by unfolding their complex structure by disrupting the covalent bonds and, thereby, increasing the availability of hydrolytic sites [129]. Therefore, by inducing different structural and microstructural modifications in the dietary proteins and food matrices, different processing technologies can affect the digestibility of food proteins differently and can have a positive or a negative influence on the gut microbiota and consequently on human health. In general, glycation and oxidation are two important chemical protein-modification reactions that occur during food processing and preparation, which play key roles in the influence of food proteins on the diversity of gut microbiota.

#### 3.5.1. Protein Glycation

Glycation (mainly through the Maillard reaction) is a common reaction that occurs during food processing [130]. Glycation can modify protein structure and produce glycated proteins [131], which lead to a decrease in protein digestibility [132]. Several studies have reported a negative impact of glycation on the digestibility of food proteins. For example, Yang et al. (2021) [133] reported a decrease in the digestibility of glycated egg white proteins, and Jiménez-Saiz et al. (2011) [134] observed a decrease in the digestibility of ovalbumin after induced glycation through the Maillard reaction. Glycation changes the protein gel structure, making it denser and more compact and reducing the diffusion of digestive enzymes. The authors concluded that glycation can be used to modify the ratio of granular and fibrous aggregates to control the gelling and digestibility properties of food products. This negative impact of glycation on protein digestibility subsequently results in more protein flow into the colon, to be fermented by gut microbiota [30]. Protein fermentation by gut microbiota can be delayed due to the glycation [30,90], but this might be due to the impact of the protein modification on the microbial population rather than the protein being resistant to microbial actions. In support of this contention, the composition of gut microbiota was found to be influenced by the extent of glycation [39]. The relative abundance of genera *Lactococcus* was markedly increased in rats fed with glycated fish protein (GFP) compared to a group of rats that were fed with fish protein (FP). Interestingly, the relative abundance of *Holdemania*, *Streptococcus*, *Enterococcus*, and *Lactobacillus* was increased upon feeding GFP heated for 48 h compared to feeding GFP heated for 24 h. This change in the microbiota population may reduce the production of ammonia and indole that have detrimental effects on the host’s health [39] (Table 1). The abundance of *Allobaculum*, *Akkermansia*, *Turicibacter*, and *Lactobacillus animalis* was increased, and the abundance of *Escherichia*-*Shigella*, *Fusobacterium*, and *Erysipelatoclostridium* was decreased in rats fed with GFP, which positively correlated with the production of butyrate [30] (Table 1). The above information indicates that the composition of gut microbiota may positively be affected by glycated proteins.

#### 3.5.2. Protein Oxidation

Protein oxidation is one of the most important chemical modifications in food proteins that occur at various levels depending on storage conditions and food processing and preparation at the industrial or household level [135,136]. There is a complex relationship between protein oxidation and digestibility. Protein hydrophobicity, solubility, and conformation could be influenced by protein oxidation [49]. Furthermore, oxidation at different temperatures affects protein conformation and denaturation differently and results in varying digestibility. For example, the cleavage sites of proteins are exposed to digestive enzymes when the proteins undergo denaturation at low temperatures [40], whereas the cleavage sites of proteins are hidden and inaccessible when protein aggregation by oxidation occurs at high temperatures [137]. Extensive oxidation is the primary cause of reduced protein digestibility, whereas gastrointestinal enzymes, such as pepsin and trypsin, perform hydrolysis more effectively on mildly oxidized and partially denatured and unfolded proteins [138,139]. Higher processing temperatures can induce intense protein oxidation and reduce the susceptibility of proteins to digestive enzymes by inducing unfavorable modifications such as the formation of cross-links and intermolecular aggregates at the protein level and the formation of disulfide linkages, amide bonds, or dityrosine bridges at amino acid levels. These changes decrease the hydrolysis of proteins and the subsequent release of amino acids [106,140], leading to the subsequent excessive accumulation of undigested proteins in the colon and subsequently influencing the composition of gut microbiota. For instance, pork oxidation induced by cooking was found to lead to higher amounts of undigested protein in the colon [40]. Mice fed with highly oxidized pork had a lower relative abundance of the *Akkermansia* (mucin-degrading bacteria), *Lactobacillus* and *Bifidobacterium* (beneficial bacteria), and *Desulfovibrio* (H_2_S-producing bacteria). Parallel to this, a higher relative abundance of *Escherichia*-*Shigella* (pro-inflammatory bacteria) and *Mucispirillum* (pathobiont) were found with the highly oxidized pork treatment (Table 1). Collectively, the changes in the microbiota were found to have a close association with damage of the intestinal barrier and to cause an inflammatory response [40] (Table 1). Given the negative impact of oxidized proteins on gut microbiota, the prevention of protein oxidation in consumed foods appears to be a key for better health.

### 3.6. Effect of Protein Structure on the Gut Microbiota

The structure of protein regulates its digestibility, the release of nutrients, and subsequent substrate availability to gut microbiota. Xie et al. (2020) [20] found that the digestibility of meat products was significantly altered by different food processing conditions (emulsion, dry-curing, stew, or steam cooked) due to their varying effects on protein structure, which led to different compositions and functions of gut microbiota. An increase in specific gut microbes (significantly associated with SCFAs) and a decrease in the relative abundance of *Akkermansia* were observed in mice fed with stewed pork compared to other protein sources [20] (Table 1). The diversity and richness of gut microbiota were lowest in rats fed with water-boiled salted duck compared to duck cooked under different styles (cooked, wine-cured, and with added sauce). The beneficial bacteria (such as *Lactobacillus*, *Allobaculum*, and *Eubacterium*) were increased in rats fed with wine-cured duck protein [141]. *Allobaculum* [142] and *Eubacterium* [143] are butyrate-producing bacteria, and *Lactobacillus* could modulate the release of anti-inflammatory components [144], which play an important role in the regulation of intestinal inflammation.

Additionally, digestibility and absorption of dietary protein were influenced by the structure of the lipoprotein emulsion [31]. Rats fed with liquid-fine emulsion (lipoprotein) had a higher relative abundance of *Parabacteroides* and a lower abundance of *Bifidobacterium*, *Sutterella*, *Parasutterella* genera, and *Clostridium Cluster XIV* compared to rats fed with gelled-coarse emulsions (lipoprotein) [31] (Table 1). The gelled-coarse lipoprotein emulsion structure resulted in a high abundance of *Akkermansia muciniphila*, *Clostridiaceae*, *Streptococcaceae*, *Bifidobacterium*, and *Clostridium Cluster XIV*. Similarly, the abundance of *Lactobacillaceae* (in the ileum) and the β-diversity of caecum mucus-associated bacteria were enhanced in rats fed with liquid-fine emulsion [32] (Table 1). *Akkermansia muciniphila* is known to demonstrate favorable functions such as strengthening the integrity of the gut barrier, enhancing gut health, and providing protection from several diseases [117], including colitis, diabetes, and obesity [118]. *Clostridium Cluster XIV* and *Bifidobacterium* are involved in metabolizing undigestible nutrients and producing beneficial metabolites such as short-chain fatty acids, which maintain intestinal homeostasis [116]. Thus, increased abundances of these families illustrate potentially beneficial outcomes associated with whey protein in the gelled-coarse lipoprotein emulsion structure. *Clostridiaceae*, which has been associated with protein metabolism, is known to consist of opportunistic pathogens [145], and an increased abundance of *Clostridiaceae* has been linked to the prevalence of arthritis in inflammatory bowel disease and rheumatoid arthritis patients [146]. *Streptococcaceae* consists of numerous opportunistic pathogens [147]. Hence, high abundances of these families postulate potentially adverse microbiota changes associated with gelled coarse lipoprotein emulsion.

## 4. Strengths and Limitations of This Review

PRISMA protocol was used as a key strategy to search and collect relevant studies to conduct a systematic review on the impact of protein on gut microbiota. While this approach resulted in a comprehensive summary of studies published from 2011 to 2020, there are some limitations in the present study. The animal subjects varied in breed, origin, age, and number among the summarized studies, and these are well-known factors that can affect the composition of gut microbiota [148,149]. Some studies had a limited number of samples due to the unavailability of intestinal digesta, injury, or death of the animal subjects during the dietary intervention. Differences in the design of the studies may also influence the impact of dietary proteins on gut microbiota. Although the language of articles was not restricted in the search strategy, papers that were not published in English were not included due to language barriers.

## 5. Conclusions

The gut microbiome is a complex system that is influenced by several factors, and its role in health and disease is of paramount importance. Analysis of the selected studies outlines the positive and negative effects of various dietary proteins and highlights the effects of some protein modifications on protein–microbiota interactions and their impact on gut microbiota. The reviewed studies suggest that diet induces rapid alterations in the gut microbial composition, and these alterations may be temporary due to the high resilience of the gut microbiota that can restore a balanced state. Protein content, its source, processing methods, and interactions with other nutrients are some of the factors that can regulate the digestion of dietary proteins and metabolites generated by gut microbiota. These entailing areas can be further investigated to expand our understanding of protein functionality and to minimize the adverse effects associated with proteolytic fermentation. Future studies should focus on evaluating the impact of dietary interventions of longer durations and monitor temporal microbiota changes to elucidate the impact of long-term consumption of various proteins on gut microbiota. Additionally, the complexity and variation in the gut microbial profiles of the individual subjects should be considered, which may lead to variations in physiological effects of different dietary proteins and differences in the risk of diseases among individuals. In this context, the adaptation of gut microbiota in individuals from different cultural and race backgrounds needs to be investigated to understand long-term colonization and potential sensitivities to protein and diet changes. Investigation into the influence of dietary protein on gut microbiota at a species level is also required to elucidate the impact of specific gut bacterial populations on host physiology due to high functional redundancy amongst various gut microbes.

## Figures and Tables

**Figure 1 nutrients-14-00453-f001:**
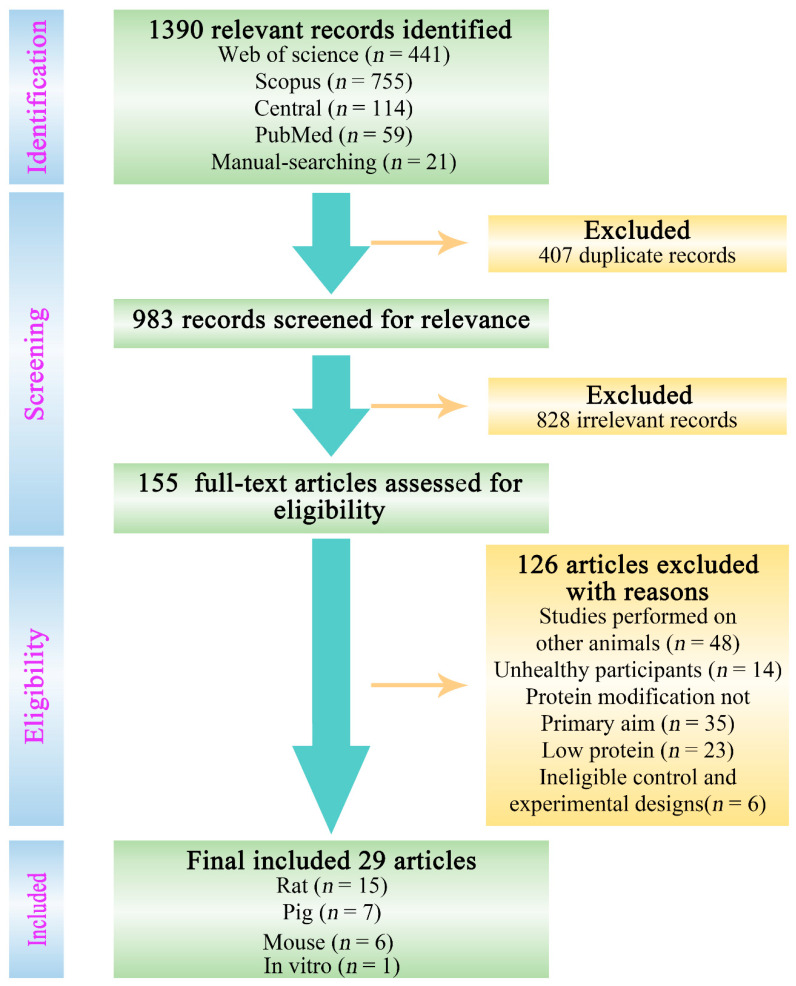
Flowchart detailing the process of identifying and selecting studies.

**Figure 2 nutrients-14-00453-f002:**
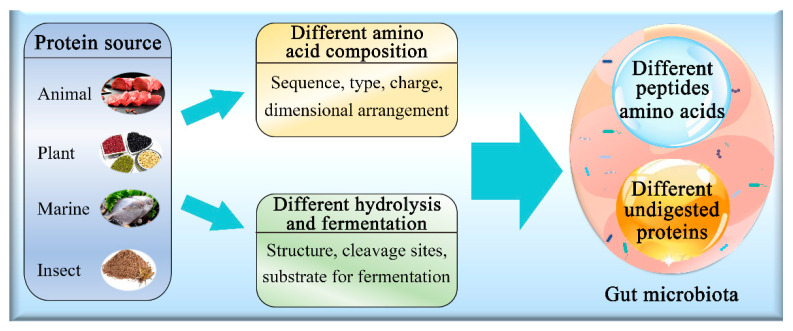
The effect of different protein sources on gut microbiota.

**Figure 3 nutrients-14-00453-f003:**
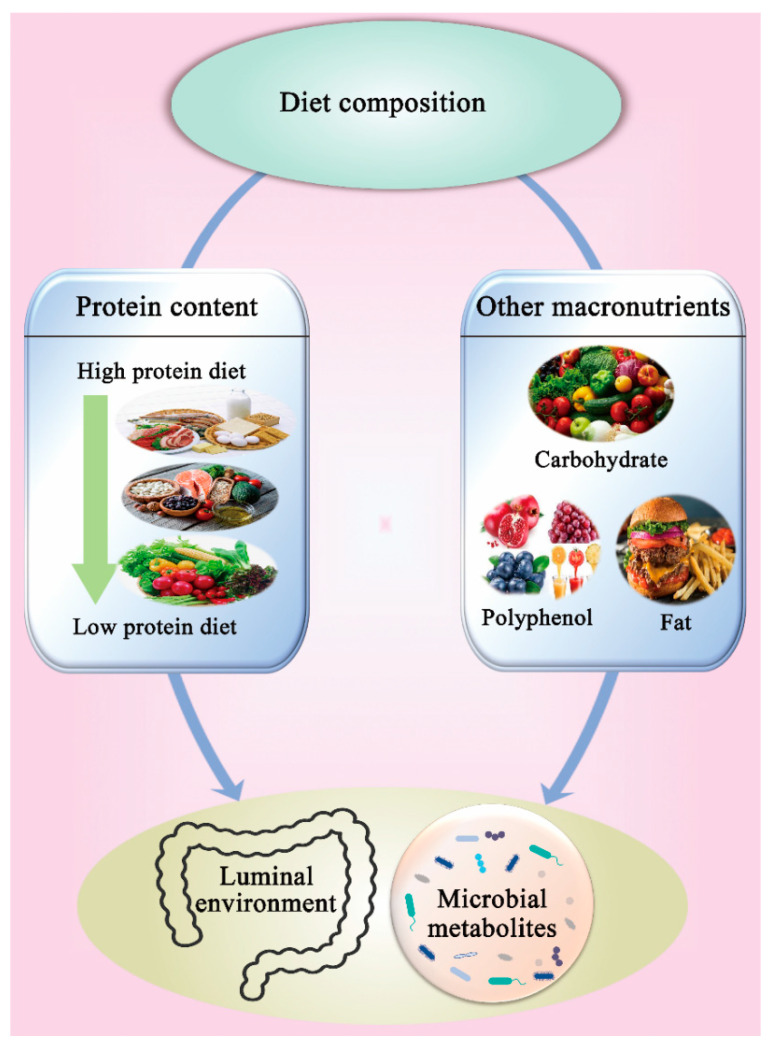
The effect of diet composition on gut microbiota.

**Figure 4 nutrients-14-00453-f004:**
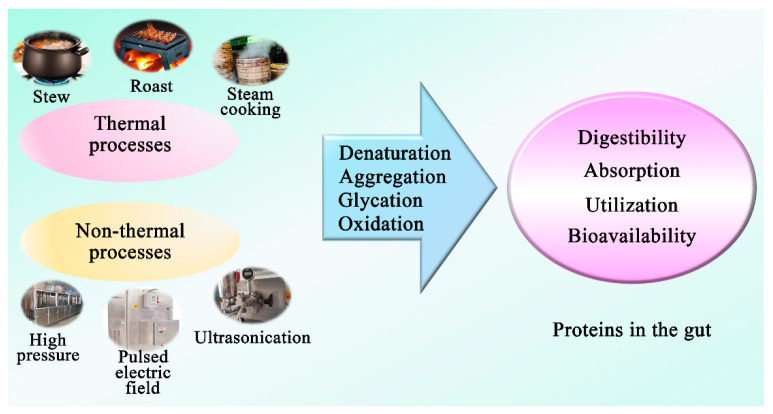
The effect of food processing on protein to influence gut microbiota.

**Table 1 nutrients-14-00453-t001:** The impact of dietary protein on gut microbiota, expressed as relative abundance (% of total microflora).

Model (Type; Gender; Age of Animals; Number of Animals Examined; Acclimation Period; Feeding Period)	Sample and Analytical Method	Protein Source (Dose)	Microbial Populations Increased: Phylum (P), Family (F), Genus (G), Species (S)	Microbial Populations Decreased: Phylum (P), Family (F), Genus (G), Species (S)	References
Mouse (Balb/c; female; 6-week-old; 4; 1 week; 2 weeks)	Feces; pyrosequencing	NP: Casein (20%) HP: Casein (30%)	Bacteroidetes (P) (HP *) *Bacteroides* (G) (HP *) *Parabacteroides* (G) (HP)	Firmicutes (P) (HP) Lachnospiraceae (F) (HP *) Ruminococcaceae (F) (HP) *Oscillibacter* (G) (HP) *Enterococcus* (G) (HP)	[9]
Rat (Wistar; male; NA ^1^; 16; 6 days; 15 days)	Cecal contents and Colonic contents; qPCR ^2^ and DGGE ^2^	NP: Whole milk proteins (14%) HP: Whole milk proteins (53%)	Cecal contents *Fusobacterium* (G) (HP)	Cecal contents and Colonic contents *Clostridium* (G) (HP) Cecal contents *Escherichia coli* (S) (HP) Colonic contents *Bifidobacterium* (G) (HP) *Fusobacterium* (G) (HP)	[15]
Piglets (Pietrain; Large White × Landrace; male and female; new-born; 48; 1 week; 2 weeks)	Colonic contents; qPCR ^2^	NP: Whey (15%) + Potassium caseinate (8%) HP: Whey (20%) + Potassium caseinate (15%)	No difference was observed in composition of the major gut microbiota.	No difference was observed in composition of the major gut microbiota.	[16]
Rat (Wistar; male; NA ^1^; 20; 1 week; 6 weeks)	Colonic contents; qPCR ^2^	NP: Casein (20%) HP: Casein (54%)	Bacteroidetes (P) (HP *) *Lawsonia* (G) (HP) *Bacteroides* (G) (HP) *Parabacteroides* (G) (HP) *Escherichia/Shigella* (G) (HP) *Enterococcus* (G) (HP) *Streptococcus* (G) (HP) *Lactobacillus* (G) (HP) *Lactococcus* (G) (HP) *Alistipes* (G) (HP) *Eubacterium* (G) (HP)	Firmicutes (P) (HP) Actinobacteria (P) (HP) Acidobacteria (P) (HP) *Sporobacter* (G) (HP) *Bifidobacterium* (G) (HP) *Ruminococcus* (G) (HP) *Akkermansia* (G) (HP) *Prevotella* (G) (HP) *Barnesiella* (G) (HP *) *Blautia* (G) (HP) *Roseburia* (G) (HP) *Allobaculum* (G) (HP) *Coprococcus* (G) (HP)	[17]
Rat (Wistar; male; NA ^1^; 20; 1 week; 6 weeks)	Feces; qPCR ^2^	NP: Casein (20%) HP: Casein (54%)	Week 1 *Lactobacillus* (G) (HP) *Bifidobacterium* (G) (HP)	Week 2 *Prevotella* (G) (HP) *Lactobacillus* (G) (HP) *Bifidobacterium* (G) (HP) Week 4 *Prevotella* (G) (HP) *Bifidobacterium* (G) (HP) Week 6 *Bifidobacterium* (G) (HP) *Prevotella* (G) (HP)	[18]
Mouse (C57BL/6J; male; 3-week-old; 36; 1 week; 4 weeks)	Ileal contents; Illumina sequencing technology ^3^	Sm: Soybean meal (30%) Ca: Casein (30%) Dw: Delactosed whey (30%) Sdp: Spray dried plasma (30%) Wgm: Wheat gluten meal (30%) Ymw: Yellow meal worm (30%)	Bacteroidetes (P) (Sm ***) Firmicutes (P) (Ca *, Dw *, Wgm *, Sdp ***, Ymw ***) Actinobacteria (P) (Ca, Sdp, Dw **) Proteobacteria (P) (Ca **) Deferribacteres (P) (Sdp **)	Firmicutes (P) (Sm **) Bacteroidetes (P) (Ca, Dw, Sdp, Wgm, Ymw **) Deferribacteres (P) (Dw **) Proteobacteria (P) (Ymw **)	[19]
Mouse (C57BL/6J; male; 4-week-old; 60; 2 weeks; 8 months)	Cecal contents; Illumina sequencing technology ^3^	Ca: Casein (20%) So: Soy (20%) Esp: Emulsion-type sausage protein (20%) Dpp: Dry-cured pork protein (20%) Spp: Stewed pork protein (20%) Cpp: Steam-cooked pork protein (20%)	Firmicutes (P) (Ca *, Dpp *, Spp *, Cpp *) Bacteroidetes (P) (So, Esp, Dpp, Spp, Cpp) Proteobacteria (P) (So, Spp, Cpp) Actinobacteria (P) (So **, Esp **) *Muribaculaceae-Norank* (G) (So, Esp, Dpp, Spp, Cpp) *Lactobacillus* (G) (So **) *Faecalibaculum* (G) (So *, Esp *, Dpp *, Cpp *, Spp ***) *Lachnospiraceae-Uncultured* (G) (Dpp **)	Bacteroidetes (P) (Ca **) Firmicutes (P) (So ****, Esp ****) Verrucomicrobia (P) (Cpp, Spp **) *Blautia* (G) (Esp, Dpp, Spp, Cpp, So **) *Akkermansia* (G) (Cpp, Spp **) *Lachnospiraceae-Uncultured* (G) (So **) *Lachnospiraceae Nk4a136* (G) (So **) *Lachnoclostridium* (G) (So **) *Ruminiclostridium* 9 (G) (So **)	[20]
Mouse (C57BL/6J; male; 5-week-old; 18; 1 week; 4 weeks)	Cecal contents and Colonic contents; Illumina sequencing technology ^3^ and qPCR ^2^	So: Soy (20%) Ch: Chicken (20%)	Firmicutes (P) (So *, Ch *) Proteobacteria (P) (So, Ch) Actinobacteria (P) (So, Ch) Verrucomicrobia (P) (Ch) *Lactobacillus* (G) (So, Ch)	Bacteroidetes (P) (So, Ch) Verrucomicrobia (P) (So) Deferribacteres (P) (So, Ch)	[21]
Rat (Sprague-Dawley; male; NA ^1^; 30; NA ^1^; 2 weeks)	Cecal; qPCR ^2^	Ca: Casein (21%) So: Soy (20%) Ze: Zein (24%)	*Lactobacillus* (G) (Ca **) *Bifidobacterium* (G) (Ca **) *Escherichia* (G) (Ze, So **)	*Escherichia* (G) * (Ca **) *Lactobacillus* (G) (So, Ze **) *Bifidobacterium* (G) (So, Ze **)	[22]
Rat (Wistar; male; 4-week-old; 18; 7 days; 16 days)	Cecal contents; pyrosequencing and DGGE ^2^	Ca: Casein (20%) So: Soy (20%) Fm: Fish meal (20%)	Firmicutes (P) (Ca *, So *, Fm *) *Turibacter* (G) (Ca **) *Oscillibacter* (G) (So **) *Lactobacillus* (G) (Fm ***)	Ruminococcaceae (F) (Ca **, Fm **) Lactobacillaceae (F) (So **, Fm **)	[23]
Rat (Sprague-Dawley; male; 3-week-old; 20; 1 week; 1 week)	Cecal; Illumina sequencing technology ^3^	Ca: Casein (20%) Ch: Chicken (20%)	Bacteroidetes (P) (Ca *) Firmicutes (P) (Ch *) Verrucomicrobia (P) (Ch) *Bacteroides* (G) * (Ca *, Ch *)	Bacteroidetes (P) (Ch *) Proteobacteria (P) (Ch) Actinobacteria (P) (Ch) *Mycobacterium* (G) (Ch) *Tetragenococcus* (G) (Ch) *Lactococcus* (G) (Ch)	[24]
Rat (Sprague-Dawley; male; NA ^1^; 66; 7 days; 90 days)	Cecal contents; Illumina sequencing technology ^3^	Ca: Casein (20%) Ch: Chicken (20%) Fi: Fish (20%) Po: Pork (20%) Be: Beef (20%) So: Soy (20%)	Firmicutes (P) (Ca *, So *, Be *, Po *, Ch ***, Fi ***) Fusobacteria (P) (Ca **) Bacteroidetes (P) (So **) *Roseburia* (G) (Ca **, So **) *Bacteroides* (G) (Ca, Be, Po, So*) *Alloprevotella* (G) (Ca, So) Proteobacteria (P) (Be **) Tenericutes (P) (Be **, Po **) *Oscillibacter* (G) (Be, Po) Actinobacteria (P) (Ch **) *Lactobacillus* (G) (Fi, Ch **)	Fusobacteria (P) (So, Be, Po, Ch, Fi) Bacteroidetes (P) (Ch **, Fi **) *Lactobacillus* (G) (Ca **)	[25]
Rat (Sprague-Dawley; male; 4-week-old; 55; 7 days; 14 days)	Feces; Illumina sequencing technology ^3^	Ca: Casein (19%) Fi: Fish (19%) Po: Pork (19%) Be: Beef (19%) So: Soy (19%)	Bacteroidetes (P) (Ca ***, So ***) Spirochaetae (P) (Ca **) Proteobacteria (P) (So **) Firmicutes (P) (Po *, Fi *, Be ***) *Bacteroides* (G) (Ca *, So ***) *Alloprevotella* (G) (Po, Be, S **) *Blautia* (G) (So **, Be **, Fi **) *Lactobacillus* (G) (Be *, Fi *, Po ***)	Spirochaetae (P) (So, Po, Be, Fi) Bacteroidetes (P) (Po, Fi, Be **) Fusobacteria (P) (Po, Be, Fi) *Lactobacillus* (G) (Ca **) *Treponema* (G) (Po, Be, Fi, So **) *Bacteroides* (G) (Be, Fi, Po **) *Fusobacterium* (G) (Po, Be, Fi) *Alloprevotella* (G) (Po **)	[26]
Rat (Sprague-Dawley; male; 3-week-old; 190; 7 days; 14 days)	Cecal contents; DGGE ^2^	Ca: Casein (20%) Ch: Chicken (20%) Fi: Fish (20%) Po: Pork (20%) Be: Beef (20%) So: Soy (20%)	Day 7 *Robinsoniella peoriensis* (S) (So, Be, Ch, Fi, Ca **, Po **) *Clostridium hathewayi* (S) (Ca **, Be **) *Blautia wexlerae* (S) (So **) Day 14 *Blautia wexlerae* (S) (So **)	Day 14 *Robinsoniella peoriensis* (S) (Ca, So, Be, Po, Ch, Fi,) *Clostridium hathewayi* (S) (Ca, So, Be, Po, Ch, Fi,) *Akkermansia muciniphila* (S) (So **)	[27]
Rat (Sprague-Dawley; male; 4-week-old, 32; 7 days; 90 days)	Colonic contents; Illumina sequencing technology ^3^	Ca: Casein (20%) Ch: Chicken (20%) Be: Beef (20%) So: Soy (20%)	Firmicutes (P) (Ca ***) Bacteroidetes (P) (So, Be, Ch **) Spirochaeta (P) (Ch, So **) Proteobacteria (P) (Be **) Tenericutes (P) (Ch **) *Lactobacillus* (G) (Ch **)	Bacteroidetes (P) (Ca **) Firmicutes (P) (So*, Be*, Ch ****)	[28]
Rat (Sprague-Dawley; male; NA ^1^; 40; 1 week; 8 weeks)	Cecal contents; Illumina sequencing technology ^3^	Ca: Casein (20%) Hew: Hen egg white (20%) Dew: Duck egg white (20%) Pew: Preserved egg white (20%)	Firmicutes (P) (Ca ***) Actinobacteria (P) (Ca **) Bacteroidetes (P) (Dew, Pew, Ca, Hew **) Verrucomicrobia (P) (Pew, Ca, Hew **) Proteobacteria (P) (Dew **) *Akkermansia* (G) (Ca, Hew **)	Bacteroidetes (P) (Ca **) Firmicutes (P) (Dew, Pew, Hew ****) Actinobacteria (P) (Dew, Pew, Hew)	[29]
Rat (Sprague-Dawley; male; NA ^1^; 32; 1 week; 2 weeks)	Cecal contents; Illumina sequencing technology ^3^	Ca: Casein (18%) FiC: Fish (12%) + Casein (6%) HfC: Heated fish (12%) + Casein (6%) GfC: Glycated fish (12%) + Casein (6%)	Firmicutes (P) (FiC *) Actinobacteria (P) (HfC, GfC **) *Lactobacillus* (G) (FiC, GfC, HfC **) *Collinsella* (G) (HfC) *Ruminococcaceae_UCG-014* (G) (GfC **) *Turicibacter* (G) (GfC **)	Bacteroidetes (P) (FiC, GfC, HfC **) Fusobacteria (P) (FiC, HfC) Proteobacteria (P) (HiC, GfC) *Allobaculum* (G) (FiC) *Fusobacterium* (G) (FiC, HfC, GfC **) *Bacteroides* (G) (HfC**, GfC **) *Subdoligranulum* (HfC**, GfC **) *Erysipelatoclostridium* (G) (GfC **) *Escherichia-Shigella* (G) (GfC **) *Ruminococcaceae_UCG-009* (G) GfC **)	[30]
Rat (Wistar Han; male; 7-week-old; 40; 1 week; 3 weeks)	Feces; Illumina sequencing technology ^3^	Lfe: Lipid-protein liquid-fine emulsions (17%) Gce: Gelled-coarse emulsions (17%)	Firmicutes (P) (Lfe *, Gfe *) Bacteroidetes (P) (Lfe, Gfe) *Parabacteroides* (G) (Lfe, Gfe) *Clostridium Cluster Xiv* (G) (Gfe) *Bifidobacterium* (G) (Gfe) *Sutterella* (G) (Gfe)	Proteobacteria (P) (Lfe, Gfe) Actinobacteria (P) (Lfe, Gfe) Deferribacteres (P) (Lfe, Gfe) Tenericutes (P) (Lfe, Gfe) *Parasutterella* (G) (Lfe, Gfe) *Clostridium Cluster Xiv* (G) (Lfe)*Bifidobacterium* (G) (Lfe) *Sutterella* (G) (Lfe) *Parasutterella* (G) (Lfe)	[31]
Rat (Wistar Han; male; 7-week-old; 16; NA ^1^; 3 weeks)	Ileal contents, Cecal contents, and Cecal mucus; Illumina sequencing technology ^3^	Lfe: Lipid-protein liquid-fine emulsions (21%) Gce: Gelled-coarse emulsions (21)	Ileal Firmicutes (P) (Lfe *, Gfe *) Proteobacteria (P) (Lfe, Gfe) *Lactobacillus* (G) (Lfe) *Bifidobacterium* (G) (Gce) Cecal Bacteroidetes (P) (Lfe *, Gfe *) Firmicutes (P) (Lfe, Gfe) *Lactobacillus* (G) (Lfe) *Coprococcus* (G) (Lfe) Verrucomicrobia (P) (Gce) *Bifidobacterium* (G) (Gce) Cecal Mucus *Coprococcus* (G) (Lfe, Gce) *Lactobacillus* (G) (Lfe) *Bifidobacterium* (G) (Gce)	Ileal Bacteroidetes (P) (Lfe, Gfe) Actinobacteria (P) (Lfe, Gfe) *Bifidobacterium* (G) (Lfe) *Lactobacillus* (G) (Gce) Cecal Proteobacteria (P) (Lfe, Gfe) Deferribacteres (P) (Lfe, Gfe) Cyanobacteria (P) (Lfe, Gfe) Verrucomicrobia (P) (Lfe) *Bifidobacterium* (G) (Lfe) *Lactobacillus* (G) (Gce) *Coprococcus* (G) (Gce) Cecal Mucus *Bifidobacterium* (G) (Lfe) *Oscillospira* (G) (Lfe)*Lactobacillus* (G) (Gce) *Coprococcus* (G) (Gce)	[32]
Piglets (Crossbred; 184 male and 152 female; 21-day-old; 336; NA ^1^; 21 days)	Ileal contents; Illumina sequencing technology ^3^	SW: Soybean meal + Whey FSW: Fish meal + Soybean meal + Whey MSSW: Microbially enhanced soybean meal + Soybean meal + Whey	Firmicutes (P) (SW ***) Proteobacteria (P) (MSSW, FSW **) Actinobacteria (P) (MSSW **)	Firmicutes (P) (FSW *, MSSW ****) Actinobacteria (P) (SMW **, FSW **) Proteobacteria (P) (SMW **)	[33]
Piglets (Crossbred; male and female; 34 days; 96; NA ^1^; 21 days)	Ileal contents; Illumina sequencing technology ^3^	SSm: 12% Soy + 9% Soybean meal FSS: 4% Fish meal + 7% Soy + 9% Soybean SHSS: Salmon protein hydrolysate (10%) + Soy (7%) + Soybean meal (9%)	Firmicutes (P) (SSm *, FSS *, SHSS *) *Lactobacillus* (G) (SSm *, FSS *, SHSS *) *Turicibacter* (G) (SSm *, FSS *, SHSS *)	Enterobacteriaceae (F) (SSm, FSS, SHSS)	[34]
Piglets (German Landrace × Piétrain; NA ^1^; 48; NA ^1^; 16 days)	Feces and Ileal contents; qPCR ^2^	Ca: Casein (10%, 16%, 22%, 27%, 33%, 39%) Sm: Soybean meal (21%, 34%, 46%, 59%, 72%, 84%)	Effect of increasing protein level: Ileal *Lactobacillus* (G) (Ca, Sm) *Bifidobacteria* (G) (Sm) Fecal *Bacteroides* (G) (Sm) *Bifidobacteria* (G) (Sm)	Effect of increasing protein level: Ileal *Clostridium Cluster XIVa* (G) (Sm) *Bacteroides* (G) (Sm) Fecal *Clostridium Cluster IV* (G) (Sm)	[35]
Piglets (German Landrace × Piétrain; NA ^1^; 48; NA^1^; 16 days)	Feces and Ileal contents; qPCR ^2^	Ca: Casein Sm: Soybean meal	Ileal *Bacteroides* (G) (Ca) *Clostridium Cluster* XIVa (G) (Ca) Fecal *Bifidobacteria* (G) (Sm) *Lactobacillus* (G) (Sm) *Bacteroides* (G) (Sm) *Clostridium Cluster IV* (G) (Sm) *Clostridium Cluster XIVa* (G) (Sm)	Ileal *Bacteroides* (G) (Sm) *Clostridium Cluster XIVa* (G) (Sm) Fecal *Bifidobacteria* (G) (Ca) *Lactobacillus* (G) (Ca) *Bacteroides* (G) (Ca) *Clostridium Cluster IV* (G) (Ca) *Clostridium Cluster XIVa* (G) (Ca)	[35]
Piglets (Crossbred; male; NA^1^; 24; NA ^1^; 10 days)	Colonic contents; Illumina sequencing technology ^3^	Dpmt: Dried porcine mucosal tissue (34%) ESm: Enzyme-treated soybean meal (35%) Cdcp: Concentrated degossypolized cottonseed protein (29%) Sdfm: Steam dried fish meal (26%)	Firmicutes (P) (Dpmt *, ESm *, Cdcp ***) Bacteroidetes (P) (Dpmt *, ESm *, Sdfm ***) Proteobacteria (P) (Dpmt **) Spirochaetes (P) (ESm, Sdfm) *Escherichia* (G) (ESm, Sdfm, Dpmt **) *Clostridium* (G) (Dpmt, Sdfm) *Campylobacter* (G) (Dpmt **) *Faecalibacterium* (G) (Dpmt **, ESm **) *Prevotella* (G) (ESm ***, FM ***) *Roseburia* (G) (Dpmt **) *Turibacter* (G) (Dpmt **) *Gemmiger* (G) (ESm) *Oscillospira* (G) (ESm **) *Lactobacillus* (G) (Cdcp **) *Megasphaera* (G) (Cdcp **) *Bacteroides* (G) (Sdfm **) *Parabacteroides* (G) (Sdfm **) *Prevotella* (G) (Sdfm ***) *Ruminococcus* (G) (Sdfm **)	Spirochaetes (P) (Dpmt **, Cdcp **) Proteobacteria (P) (Sdfm, ESm, Cdcp **) Bacteroidetes (P) (Cdcp **) Firmicutes (P) (Sdfm **) *Ruminococcus* (G) (Dpmt **, SBM **) *Prevotella* (G) (ESm ****, CDCP ****) *Roseburia* (G) (Cdcp **) *Phascolarctobacterium* (G) (Cdcp **) *Roseburia* (G) (Cdcp **)	[36]
Piglets (Duroc × Landrace × Large White; female; NA ^1^; 72; NA ^1^; 46 days)	Colonic contents; Illumina sequencing technology ^3^ and qPCR ^2^	Sm: Soybean meal (17%) H1Sm: *Hermetia illucens* larvae (4%) + Soybean meal (14%) H2Sm: Hermetia illucens larvae (8%) + Soybean meal (11%)	Firmicutes (P) (Sm *, H1Sm *, H2Sm ***) Bacteroidetes (P) (Sm **, H1Sm **) Actinobacteria (P) (H1Sm **, H2Sm **) Proteobacteria (P) (H2Sm **) *Bacteroides* (G) (SM ** *Pseudobutyrivibrio* (G) (H1Sm) *Oribacterium* (G) (H1Sm) *Lactobacillus* (G) (H2Sm, H1Sm **) *Roseburia* (G) (H2Sm, H1Sm **) *Faecalibacterium* (G) (H2Sm, H1Sm **) *Clostridium cluster IV* (G) (H2Sm, H1Sm)	*Bacteroidetes* (P) (H2Sm **) *Streptococcus* (G) (H2Sm, H1Sm **) *Treponema* (G) (H2Sm, H1Sm **) *Bacteroides* (G) (H2Sm, H1Sm **) *Eubacterium* (G) (H1Sm **) *Barnesiella* (G) (H1Sm) *Oscillibacter* (G) (H1Sm)	[37]
Piglets (Crossbred; female; 13-week-old; 45; NA ^1^; 4 weeks)	Feces; qPCR ^2^	Ca: Casein (13–15%) Lu: Lupin (13–15%) Be: Beef (13–15%)	Proteobacteria (P) (Ca *, Be ***) Actinobacteria (P) (Ca **, Lu **) Firmicutes (P) (Lu ***) Bacteroidetes (P) (Lu **)	Bacteroidetes (P) (Ca *, Be ***) Firmicutes (P) (Ca *, Be ***) Proteobacteria (P) (Lu *) Actinobacteria (P) (Be **)	[38]
In vitro batch fermentation model of human distal colon	-	Fp: Fish protein HFp24: Heated (24 h) fish protein HFp48: Heated (48 h) fish protein GFp24: Glycated (24 h) fish protein GFp48: Glycated (48 h) fish protein	Fusobacteria (P) (FP *, HFp24 *, HFp48 *, GFp48 *) Bacteroidetes (P) (Fp **, HFp24 **) Proteobacteria (P) (Fp, HFp24, HFp48, GFp24, GFp48) Firmicutes (P) (HFp48) *Clostridium_sensu_stricto_1* (G) (HFp48 **) *Streptococcus* (G) (HFp48 **) *Arcobacter* (G) (HFP48 **) *Holdemania* (G) (GFP48 **)	Proteobacteria (P) (Fp **) Firmicutes (P) (Fp **, HFP **) Bacteroidetes (P) (HFp24)	[39]
Mouse (Balb/c; male; 4-week-old; 30; 1 week; 12 weeks)	Cecal contents; Illumina sequencing technology ^3^	HOP: Low-oxidative damage pork MOP: Medium-oxidative damage pork LOP: High-oxidative damage pork	*Escherichia-Shigella* (G) (HOP) *Mucispirillum* (G) (HOP)	*Lactobacillus* (G) (HOP) *Bifidobacterium* (G) (HOP) *Desulfovibrio* (G) (HOP)	[40]
Rat (Sprague-Dawley; male; NA ^1^; 40; 5 days; 21 days)	Colonic contents; Illumina sequencing technology ^3^	FC: Fresh chicken CC: Cured chicken FB: Fresh beef CB: Cured beef	Ruminococcaceae (P) (CC, CB) *Oscillibacter* (G) (CB)	*Marvinbryantia* (G) (CC)	[41]
Mouse (C57BL/6J; male; 6 or 8-week-old; 20; NA ^1^; 24 weeks)	Cecal contents; Illumina sequencing technology ^3^	NP: Casein (20%) HP: Casein (52%)	Actinobacteria (P) (HP) *Bifidobacterium* (G) (HP) *Bacteroides* (G) (HP) *Parabacteroides* (G) (HP) *Oscillospira* (G) (HP)	Saccharibacteria (P) (HP)	[42]

^1^ NA = not available. ^2^ qPCR, quantitative polymerase chain reaction; DGGE, denaturing gradient gel electrophoresis. ^3^ Illumina sequencing technology, a high-throughput sequencing analysis. * the most abundant population in that diet group; ** the most or least abundant population in that diet group compared to all other diet groups; *** the most abundant population in that diet group, as well as compared to other diet groups; **** the least abundant population compared to other diet groups but the most abundant population in that diet group. The diets used within each study are balanced for all nutrients, and only protein was the study’s independent variable.

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
