# Peer review of "Effect of Dietary Protein and Processing on Gut Microbiota—A Systematic Review"

_nutrients, 2022, doi:10.3390/nu14030453_

Round 1

Reviewer 1 Report

General comment:

This review provides information of dietary protein sources on microflora. However, this review has several fundamental problems.  

Specific comments:

  1. It is unclear whether the results of microflora in Table 1 are expressed as relative abundance (% of total microflora), or absolute counts (copy numbers) of each bacterium (counts per gram of fresh matter or dry matter, or counts per total matter). Analytical methods such as qPCR, gene sequencing analysis of 16S rRNA, etc. are also unclear. Authors should be always careful in the alterations in the amounts of gut contents because several prebiotics can modulate the weight of gut contents as well as the composition of microbiota.  
  2. Further information such as the age of animals, feeding period of test diets, sample size, etc. should be provided.
  3. Since this review aims to delineate the impact of protein sources and their processing on microbiota, the discussion on the limited studies on the effects of dietary protein level on microflora should be removed.
  4. The authors must carefully check the experimental designs of the papers cited. The papers such as No.22, No.34, and No. 41 appear to be inappropriate for the citation.
  5. In general, plant proteins such as soy protein are believed to be protective against several chronic diseases. Authors should discuss the possibility of the involvement of microflora in the beneficial role of soy protein.
  6. How about the discussion on the roles of amino acids in dietary protein sources in gut microflora? Accumulating evidence indicates dietary intake of some amino acids has a key role in microflora.
  7. There are numerical problems in the references. The references were incomplete, and inconsistent.

Author Response

On behalf of my co-authors, thank you very much for you giving us the opportunity to revise our manuscript. The authors appreciate the corrections as well as constructive and thoughtful comments and suggestions provided by the reviewers and editorial office. The manuscript has been revised carefully, and all the comments have been considered. The changes are shown in different font color in the revised manuscript.

Reviewer: 1

This review provides information of dietary protein sources on microflora. However, this review has several fundamental problems.

Specific comments:

  1. It is unclear whether the results of microflora in Table 1 are expressed as relative abundance (% of total microflora), or absolute counts (copy numbers) of each bacterium (counts per gram of fresh matter or dry matter, or counts per total matter). Analytical methods such as qPCR, gene sequencing analysis of 16S rRNA, etc. are also unclear. Authors should be always careful in the alterations in the amounts of gut contents because several prebiotics can modulate the weight of gut contents as well as the composition of microbiota.

Response        Thank you for your suggestion. The results of microflora are expressed as relative abundance (% of total microflora). The full names of the standard methods have been included as footnote. The diet in the reported studies is balanced for all nutrients, except for the protein which was the independent variable. The information has been included in Table 1 (in blue color).

  1. Further information such as the age of animals, feeding period of test diets, sample size, etc. should be provided.

Response        Thank you for your suggestion. The corresponding information have been added in Table 1, including type, gender, age of animals, sample size, acclimation period, and feeding period of model and analytical sequencing method (Illumina, quantitative polymerase chain reaction, pyrosequencing and denaturing gradient gel electrophoresis). New Table 1 has been embedded at the end of this file.

  1. Since this review aims to delineate the impact of protein sources and their processing on microbiota, the discussion on the limited studies on the effects of dietary protein level on microflora should be removed.

Response        Thank you for your comments. The authors consider that section is an important part of this review since it highlight the limitations available in literature and could potentially direct future research activities to address these limitations. Therefore, we have added more information in the manuscript to further discuss it.

Revised Version   Page 2:

L 54-55: Protein intake in terms of quantity and quality is central to the above-mentioned effects with complex mechanisms involved.

L 65-68: This systematic review aims to delineate the impact of various dietary protein levels and dietary protein sources, and their processing on relative abundances of gut microbial population and examine potential underlying factors that may influence this relationship.

L 94-98: Studies were included if they met all the following criteria: (1) performed experimental research (dietary interventions/treatments) on healthy humans, mice, rats, pigs, or in-vitro; (2) dietary interventions or experimental research with protein modification was the primary aim; (3) dietary interventions that administered normal or high protein doses or increasing levels of protein;

  1. The authors must carefully check the experimental designs of the papers cited. The papers such as No.22, No.34, and No. 41 appear to be inappropriate for the citation.

Studies existing experimental designs.

Response        Thank you for your suggestion. These references have been deleted. The other references have been checked again.

  1. In general, plant proteins such as soy protein are believed to be protective against several chronic diseases. Authors should discuss the possibility of the involvement of microflora in the beneficial role of soy protein.

Response        Thank you for your suggestions. The discussion of soy protein and gut microbiota has been added.

Revised Version   Page 17:

L 213-217: This may be related to the reported digestibility differences among proteins. For exam-ple, studies have demonstrated that casein and whey proteins have different digestion kinetics (based in leucine kinetics modification) [56], beef and chicken proteins have a higher digestion rate (digestibility) than fish proteins [57] and soy proteins to have a higher digestion kinetics than milk proteins (based on nitrogen absorption, splanchnic uptake and metabolism) [58].

L 220-222: the abundance of Bacteroidales family S24-7 was enhanced in mice fed with soybean meal as the protein source compared to casein, spray dried plasma protein, yellow meal worm, partially delactosed whey powder or wheat gluten meal [19] (Table 1).

L 227-230: Bacteroidetes, a phylum that contains large number of microorganisms involved in metabolic processes involving hydrolysis of polysaccharides and proteins, were in-creased in rats fed with soy protein and decreased in rats fed with proteins from fish (Table 1).

L 233-236: An et al. (2014) [23] found higher contents of n-butyric acid, lactic acid and other putrefactive compounds in rats fed on soy protein and fish meal compared to casein, which suggested differential metabolism by gut microbiota that can lead to physiological changes in the gut.

Page 17-18:

L 240-251: Dietary proteins from animal sources and plant sources have been widely explored in relation to the modulation of gut microbiome. Many studies have reported that plant proteins (rice, soy, wheat, etc.) can improve the composition of gut microbiota [19, 61, 62]. For instance, soybeans, an important source of plant protein, have been gaining wide popularity due to their health-promoting effects [63]. Soy proteins are considered a rich of all essential amino acids that preferentially support the growth of some gut microbiota as both nutrient and energy sources [62, 63]. Soy proteins/peptides appears to modulate gut microbiota by exerting probiotic effects by enhancing probiotics (Lactobacilli and Bifidobacteria) and decreasing Bacteroidetes [62, 63]. Han et al. reported that the diversity and richness of the gut microbiota in mice were changed by fermented soy whey resulting in the enhance of Bifidobacterium, Lactobacillus, Butyricicoccus, Parabacteroides, Lachnospiraceae and Akkermansia muciniphila to affect the metabolism and health of mice [64]

  1. How about the discussion on the roles of amino acids in dietary protein sources in gut microflora? Accumulating evidence indicates dietary intake of some amino acids has a key role in microflora.

Response        Thank you for your suggestions. Discussion of amino acids and gut microbiota has been added.

Revised Version   Page 16-17:

L 186-199: In addition, growing evidence indicated that amino acids, products of dietary protein digestion, could affect the structure, composition, and functionality of gut microbiota [48, 51]. The amino acids can further be metabolized into different microbial metabo-lites by gut microbiota, such as SCFAs, polyamine, hydrogen sulfate, phenol, and in-dole, and the resultant metabolites can be involved in various physiological functions which related to host health and diseases, [48]. For example, increase in the abundanc-es of Escherichia-Shigella, Aquabacterium, and Candidatus Methylomirabilis, and decrease in the abundances of Bacteroides, Bacillus, Pasteurella, Clostridium sensu stricto, Faecali-bacterium, Paucisalibacillus, and Lachnoclostridium were found in pigs with dietary ly-sine restriction (30%), which resulted in restricted amino acid metabolism [52]. The role of amino acids in regulating the host health was supported in various studies. For example, the increase in the SCFA-producing bacteria (Bifidobacterium, Lactobacillus, Bacteroides, Roseburia, Coprococcus, and Ruminococcus) and inflammation-inhibiting bacteria (Oscillospira and Corynebacterium) as well as the decrease of inflamma-tion-causing bacteria (Desulfovibrio) were observed in mice with methionine-restricted diets, can collectively improve the gut health [53].

  1. There are numerical problems in the references. The references were incomplete, and inconsistent.

Response        Thank you for your observation. The references have been revised and all the references are complete now. Some of the online and open access articles may not have complete numbering system.

Reviewer 2 Report

It is an extended analysis on the effect of diet, and more specifically of dietary protein on the composition of gut microbiome. The article is very well documented and scrupulous written. 

Author Response

Response        Thank you for your time and positive comments.

Reviewer 3 Report

Over nine years, studies on the subject are still being carried out experimentally, most of the articles are studies involving experimental animals. There is only 1 study involving humans. Most studies are involving animals.

Authors should have also included works published in 2021. 

Author Response

On behalf of my co-authors, thank you very much for you giving us the opportunity to revise our manuscript. The authors appreciate the corrections as well as constructive and thoughtful comments and suggestions provided by the reviewers and editorial office. The manuscript has been revised carefully, and all the comments have been considered. The changes are shown in different font color in the revised manuscript.

Reviewer: 3

Authors should have also included works published in 2021.

Response        Thank you for your suggestion. The appropriate works in 2021 have been included in Table 1 (in red color) and in the main text.

Original Version   Page 3:

L 130-146: The process of the literature selection is shown in Figure 1. A total of 1361 records were identified by the literature search from the following electronic databases and the retrieved studies are shown in brackets: PubMed (n = 58), Scopus (n = 750), Web of science (core collection) (n = 439), Central (Cochrane central register of controlled trials) (n = 114), and from manual-searching of reference lists (n = 18). A total of 405 duplicate records were removed by the de-duplication process and the remaining 973 records were evaluated for relevance, which subsequently resulted in 828 records were deemed irrelevant according to the study selection criteria. We excluded studies that was conducted on other animals (e.g., cats, dogs, marine animals and insects) (47), studies investigated dietary interventions on unhealthy participants with an acute or chronic disease/condition (11), studies reporting on protein effects not being the primary aim (35), studies which focused on low protein or protein deficient/restricted diets (22), and studies that implemented ineligible control (fibre) (1). A total of thirty studies were found to meet all the set criteria and were selected to be included in this systematic review. These studies included six mouse studies, seven pig studies, fifteen rat studies, one human study and one in vitro study, which have been discussed in detail in subsequent sections. The bibliometric information of these thirty studies can be found in Table 1.

Figure 1

Revised Version   Page 3:

L 130-146: The process of the literature selection is shown in Figure 1. A total of 1390 relevant records were identified and selected by the literature search from the following electronic databases and the retrieved studies are shown in brackets: PubMed (n = 59), Scopus (n = 755), Web of science (core collection) (n = 441), Central (Cochrane central register of controlled trials) (n = 114), and from manual-searching of reference lists (n = 21). A total of 407 duplicate records were removed by the de-duplication process and the remaining 983 records were evaluated for relevance, which subsequently resulted in 828 records were deemed irrelevant according to the study selection criteria. We excluded studies that was conducted on other animals (e.g., cats, dogs, marine animals and insects) (48), studies investigated dietary interventions on unhealthy participants with an acute or chronic disease/condition (14), studies reporting on protein effects not being the primary aim (35), studies which focused on low protein or protein deficient/restricted diets (23), and studies that implemented ineligible control (fibre) and experimental designs (6). A total of twenty-nine studies were found to meet all the set criteria and were selected to be included in this systematic review. These studies included six mouse studies, seven pig studies, fifteen rat studies and one in vitro study, which have been discussed in detail in subsequent sections. The bibliometric information of these thirty studies can be found in Table 1.

Figure 1

Round 2

Reviewer 1 Report

This manuscript was well improved according to the suggestions of reviewer.

However, further improvement is necessary as below.

“illumina”

What is “illumina” ?  “illumina high-throughput sequencing analysis” ?  Explanation of “illumina” should be added to the bottom of Table 1 (p15) as well as that of qPCR.

“Sample size”

Sample size is usually provided as the number of animals examined per one group. Are the data of sample size indicated in Table 1 the numbers of animals examined per one group or the numbers of all the animals examined (for all the groups)?  Please carefully check the meaning of sample size referring  to the textbook of statistics.

Author Response

On behalf of my co-authors, I would like to thank you for you giving us the opportunity to revise our manuscript. The authors appreciate the comments received from the reviewer and editors and we thank them for their corrections, and constructive and thoughtful comments. The manuscript was carefully revised, and all the comments have been considered. The changes are shown in red color font in the revised manuscript.

Reviewer: 1

This manuscript was well improved according to the suggestions of reviewer.

However, further improvement is necessary as below.

“illumina”

What is “illumina”? “Illumine high-throughput sequencing analysis”? Explanation of “illumina” should be added to the bottom of Table 1 (p15) as well as that of qPCR.

Response        Thank you for your suggestion. The “illumina” has been replaced by “Illumina sequencing technology, a high-throughput sequencing analysis” in Table 1 (with red color). The explanation of “qPCR” has been added to the bottom of Table 1.

Revised Version   Page 15:

L 151: qPCR, quantitative polymerase chain reaction; DGGE, denaturing gradient gel electrophoresis. Illumina sequencing technology, a high-throughput sequencing analysis

“Sample size”

Sample size is usually provided as the number of animals examined per one group. Are the data of sample size indicated in Table 1 the numbers of animals examined per one group or the numbers of all the animals examined (for all the groups)? Please carefully check the meaning of sample size referring to the textbook of statistics.

Response        Thank you for your suggestion. In order to avoid misunderstanding, the “Sample size” has been replaced by “numbers of animals examined” in the Table 1.

Reviewer 3 Report

The authors have substantially changed the format of the manuscript.

Author Response

Thank you for your comments